# Empowering Community Health Workers in Japan: Determinants of Non-Communicable Disease Prevention Competency

**DOI:** 10.3390/healthcare12030297

**Published:** 2024-01-24

**Authors:** Yuki Imamatsu, Yuka Iwata, Ayuka Yokoyama, Yuko Tanaka, Etsuko Tadaka

**Affiliations:** 1Department of Community Health Nursing, Graduate School of Medicine, Yokohama City University, Fukuura 3-9, Kanazawa-ku, Yokohama 236-0004, Japan; iwata.yuk.go@yokohama-cu.ac.jp; 2Department of Nursing Informatics, Graduate School of Nursing Science, St. Luke’s International University, 10-1 Akashi-cho, Chuo-ku, Tokyo 104-0044, Japan; 22mn031@slcn.ac.jp; 3Department of Community Health Nursing, School of Nursing and Social Services, Health Sciences University of Hokkaido, 1757 Kanazawa, Tobetsu-cho, Ishikari-gun 061-0293, Japan; yta@hoku-iryo-u.ac.jp; 4Department of Community and Public Health Nursing, Graduate School of Health Sciences, Hokkaido University, K12-N5, Kita-ku, Sapporo 060-0812, Japan

**Keywords:** community health workers, competency, empowerment, prevention of non-communicable diseases, health literacy, sense of community

## Abstract

Background: Community health workers (CHWs), hailing from the general populace, play a pivotal role in fortifying healthcare systems, with a primary focus on mitigating non-communicable diseases (NCDs) and elevating overall life expectancy. To assess the aptitude of CHWs in NCD prevention, we introduced the Community Health Workers Perceptual and Behavioral Competency Scale for preventing non-communicable diseases (COCS-N). This study examines the multifaceted interplay of individual and community factors that influence CHWs’ COCS-N scores. Methods: The research design is a secondary analysis using data from a self-administered questionnaire survey of 6480 CHWs residing in municipalities across Japan, which obtained 3120 valid responses, between September to November 2020. The COCS-N was employed as the dependent variable, while the independent variables were individual-related factors, including years of community health work, health literacy, and community-related factors, such as CHWs’ sense of community. To ascertain the significance of associations between individual and community factors and CHWs’ competency, an analysis of covariance (ANCOVA) was utilized to compare the three groups Q1/Q2/Q3 by low, medium, and high scores on the COCS-N scale. Statistical significance was considered to be indicated by a *p*-value of less than 0.05. Results: The ANCOVA analysis revealed that three factors were significantly linked to CHWs’ competence. These comprised individual factors: “years of CHWs” (mean ± SD Q1: 6.0 ± 6.0, Q2: 7.8 ± 7.0, Q3: 8.2 ± 7.7, *p* < 0.001) and “health literacy” (Q1: 27.7 ± 6.6, Q2: 30.4 ± 6.9, Q3: 33.8 ± 7.8, *p* < 0.001), as well as a community factor: “Sense of community” (Q1: 14.8 ± 3.7, Q2: 16.5 ± 3.5, Q3: 18.2 ± 3.6, *p* < 0.001). Conclusions: Our finding is that a positive association was derived between COCS-N scores and certain determinants. Notably, “years of CHWs” and “health literacy” in the individual domain, along with the “Sense of community” in the communal context, were firmly established as being significantly associated with CHWs’ competency. Consequently, CHWs need training to increase their “health literacy” and “sense of community”, to acquire high competency in NCD prevention, which will lead to the empowerment of CHWs and maintain their motivation to continue.

## 1. Introduction

Community Health Workers (CHWs) are defined as local residents who are elected to provide primary health care on behalf of professional healthcare providers. These individuals are typically chosen by their communities and are accountable to them for their activities. They operate within a healthcare system established by a national or municipal public health organization, such as a ministry of health or health center [1]. CHWs play four primary roles and functions, which include health education, social support, advocacy, and coordination and mediation. Health education aims to enhance the knowledge of patients and community members, reducing diseases and their primary risk factors. Social support encompasses emotional, evaluative, informational, instrumental, and material assistance. Advocacy, coordination, and mediation assist residents in accessing health facilities and local health professionals, with CHWs acting as collaborative bridges [2,3]. The presence of CHWs significantly improves the health of large populations without imposing an undue burden. Nevertheless, for CHWs to effectively address health-related issues within their communities, the quality of their activities must be ensured. Challenges in this regard include the recruitment, training, roles, remuneration, knowledge security, and professional development of CHWs [4]. Furthermore, the most pressing challenges regarding CHWs encompass equitable remuneration, the integration of CHW activities into the health system without disrupting their community roles, combining CHWs’ contributions with other forms of social support, and achieving a favorable return on investment while yielding substantial improvements [4]. Notably, there are two types of CHWs: those employed and compensated by national and municipal health organizations, and those who serve voluntarily. Determining appropriate compensation for their efforts is a significant challenge [5].

When discussing CHWs, it is crucial to consider the diverse range of activities they undertake based on their locations and the diseases they target [4,6]. Previous studies on CHWs have shown that developing countries often train them to address infectious diseases and maternal and child health, while developed countries focus on non-communicable diseases (NCDs) like heart disease, various cerebrovascular diseases, and diabetes prevention and management [7,8,9,10,11,12]. NCDs, also known as chronic diseases or lifestyle-related diseases, encompass conditions such as cancer, diabetes, cardiovascular disease, respiratory disease, and mental illness. These diseases result from various factors, including an unhealthy diet, lack of exercise, smoking, excessive alcohol consumption, and air pollution. NCDs are characterized by their long duration and are influenced by a combination of genetic, physiological, environmental, and behavioral factors [13,14]. In 2016, international professional bodies reported staggering numbers of people affected by NCDs, with 523 million individuals having cardiovascular disease, 463 million with diabetes, and 40 million stroke survivors worldwide [15,16,17]. The incidence of cardiovascular disease had doubled in the preceding 30 years, and diabetes had more than tripled in the previous decade [15,17]. Given the global surge in NCD-related morbidity and mortality, taking preventive measures at the national level is imperative. Some studies have found that the outcomes of using CHWs for diabetes management are comparable to those of nursing professionals [18]. Other studies have developed models for the CHW workforce in primary healthcare systems [19].

In Japan, heart disease and cerebrovascular disease rank as the second and fifth leading causes of death, respectively, and NCDs remain the primary cause of death overall in other developed countries [20]. To address NCDs, Japan introduced the concept of metabolic syndrome, which combines obesity and abnormalities in cardiovascular blood test data such as blood pressure, blood glucose levels, and lipid abnormalities. Specific check-ups and health guidance were implemented to enable the early detection and treatment of individuals at high risk of NCDs [21]. However, NCD prevention efforts must target both high-risk individuals and the general population. The Ministry of Health, Labor and Welfare (MHLW) established the Health Japan 21 initiative, which aims to improve lifestyle-related diseases and habits to increase healthy life expectancy and reduce health disparities among the entire Japanese population [22]. Public health nurses with specialized knowledge in community assessment, legislation, and social resources, along with voluntary CHWs from local communities, collaborate to deliver primary health care in a unique community health system in Japan [23]. Japan’s community health care history and system, coupled with the world’s longest life expectancy at a low cost among developed nations, have successfully provided universal health coverage [24,25]. The contributions of public health nurses and CHWs to this achievement are considered meaningful activities that positively impact global health. CHWs cover the intersection between local residents and local government, and offer activities at the individual, interpersonal, group, and community levels. For example, the role of CHWs can range from introducing recipes for low-salt and low-sugar meals to prevent NCDs, to explaining how to perform weight training at home, to introducing health programs at health centers, such as health screening in communities with high blood pressure and diabetes. The most important competence among CHWs is the ability to help community members review and improve their health perceptions and behaviors on the basis of basic knowledge and skills about the prevention of NCDs [26,27].

These roles of CHWs are related to the concept of empowerment. Empowerment has long been the focus of research on patients, women, and other socially vulnerable groups [28,29]. Since the importance of empowerment was pointed out in the Alma Ata Declaration, it has also become a core concept in health-promotion activities. The concept of empowerment involves multiple levels, including community empowerment as a multilevel construct. Empowerment has been conceptualized as a process with three levels, starting with individual development and psychological empowerment at the individual level, followed by community empowerment, then social and/or political empowerment [30]. However, the various definitions of empowerment are diverse, and measurement tools are not standardized [30,31]. This will lead to the consideration of measures to empower CHWs to undertake health-promotion activities by linking their own health promotion to the ripple effects on their communities. Therefore, we defined empowerment as a process in which individual empowerment exhibits spillover effects, and leads to community empowerment; we believe that increasing the competencies of CHWs will contribute to individual and community empowerment.

To address the challenges faced by CHWs, including recruitment, training, roles, remuneration, knowledge security, and professional development, we developed the Community Health Workers Perceptual and Behavioral Competency Scale for Preventing Non-communicable Diseases (COCS-N) in Japan [27]. Competency theory distinguishes between visible competencies like knowledge and skills and hidden competencies such as beliefs and values [32,33]. In this study, competencies are defined as mechanisms that enable productive workers to achieve results, and include behavioral (e.g., skills and knowledge) and perceptual (e.g., attitudes and values) aspects [27]. As a result, COCS-N incorporates perceptual and behavioral subscales. The perceptual subscale revolves around “Sharing the pleasure of living a healthy life”, while the behavioral sub-scale focuses on “Creating healthy resources”. CHWs operate at the intersection of healthcare and community, which guided the development of COCS-N. Our goal is to address the challenges faced by CHWs, including equitable remuneration, maintaining their community roles, keeping them motivated, and effectively integrating them into the community care system alongside other physical and social support providers. Notably, CHWs in Japan are volunteers who do not receive monetary compensation, making the findings of this paper relevant to volunteer CHWs. To achieve this, we must identify the factors influencing CHWs’ competence in NCD prevention. This study examines individual- and community-related factors affecting CHWs’ COCS-N scores, aiming to uncover the associations between competence in preventing NCDs and various modifiable physical, mental, and social characteristics of CHWs, as well as structural and systematic elements within their community activities. This will lead to the consideration of measures to enhance the power, or empowerment, of CHWs to promote health-promotion activities by linking their own health promotion to the ripple effects on their communities.

## 2. Materials and Methods

### 2.1. Design, and Setting of the Study

We conducted a cross-sectional design as a secondary analysis using data from a previous study of 6480 CHWs registered in all municipalities (cities, towns, and villages) across Japan between September to November 2020 [27]. 

### 2.2. Participants

We used a maximum sample size that adhered to the criteria for the secondary analysis study, drawn from the total population of the primary study [27]. Before sending the survey questionnaires, the author sent informed consent letters to all 1743 municipalities in Japan. All 194 cities and towns (response rate: 11.1%) consented to participate in this study. These 194 cities (local government) were employers of CHWs. The inclusion criteria for the subjects of this study were as follows: (1) individuals who had engaged in public health-promotion activities as CHWs for a minimum of one year, and (2) individuals approved by their respective municipalities as being in a healthy condition to participate in this study. A total of 3651 individuals were included in this study (response rate: 56.3%). A further 531 individuals were excluded as follows: (1) CHWs whose responses to the questionnaire revealed insufficient participation in health-promotion activities (i.e., in response to the question asking how long they had been active as CHWs, they answered ‘Less than one year’, despite having received the questionnaire on the understanding that they had been active for at least one year (n = 17)), (2) CHWs who did not respond to the COCS-N or community commitment scale (CCS) items (n = 268), and (3) CHWs who failed to respond to at least two items in the European Health Literacy Survey Questionnaire (HLS-EU-Q47, n = 246). This left 3120 participants for inclusion in the analysis, with a valid response rate of 48.1%. The representativeness of this sample was confirmed in a previous study [27].

The sample size for the analysis of covariance was calculated using G*Power 3.1.2 on the basis of a moderate effect size of 0.15 (medium), and a significance level of 0.05, power of 80%, and 3 of groups and 5 predictors. Thus, the required sample size was calculated to be 731. The final sample size exceeded the pre-study sample size calculation.

### 2.3. Measurements

#### 2.3.1. Demographic Characteristics

The participants’ demographic characteristics that we measured included age, sex, family structure (living alone/with a spouse/with children/with a spouse and children/with children and grandchildren/other), years of residence, and education (primary school and below/junior high school/senior high school/vocational school or technical college/university/graduate school). 

#### 2.3.2. Dependent Variable

The Community Health Workers Perceptual and Behavioral Competency Scale for Preventing Non-communicable diseases (COCS-N) is an instrument used to assess competence in the prevention of NCDs among CHWs [27]. This scale consists of 8 items measuring perceptual competence and behavioral competence on a 4-point Likert scale as follows: 0 = not applicable, 1 = somewhat inapplicable, 2 = somewhat applicable, and 3 = entirely applicable. The total score ranges from 0 to 24. A high score indicates a high degree of competence in the prevention of NCDs. COCS-N has a Cronbach’s alpha, which conveys the internal consistency of the scale, of 0.86 in Japan. To separate the high and low COCS-N scores, we divided them into three groups according to their mean values, with reference to previous studies [34,35]. 

#### 2.3.3. Independent Variables 

##### Individual Factors

We assumed that individual factors included years of working as CHWs, health literacy, a subjective sense of health, and health status (number and type of diseases under treatment). 

Years of working as CHWs were measured by the participants’ responses to the question “How long have you been working as a CHW?” 

Health literacy was measured using the HLS-EU-Q47 (Japanese version) [36,37]. This scale consists of 12 dimensions, consisting of a total of 47 items measuring the 4 information-related competencies of health literacy (obtaining, understanding, evaluating, and using) across 3 subdomains (Health Care = HLS-EU-Q47-HC, Disease Prevention = HLS-EU-Q47-DP, and Health Promotion = HLS-EU-Q47-HP), and was measured on a 4-point Likert scale, as follows: 1 = very difficult, 2 = somewhat difficult, 3 = somewhat easy, and 4 = very easy. The total score ranges from 0 to 50; the formula for the standardized score was (mean − 1) × (50/3). A high score indicates a high degree of health literacy. In this study, scores were recorded for the overall HLS-EU-Q47 and the subdomains HLS-EU-Q47-HC, HLS-EU-Q47-DP, and HLS-EU-Q47-HP. 

A subjective sense of health was measured by participants’ responses to the question “How do you rate your health status?” on a 4-point Likert scale, as follows: 1 = very healthy, 2 = reasonably healthy, 3 = reasonably unhealthy, and 4 = very unhealthy. Participants who responded with either 1 or 2 were defined as healthy, while those who responded with either 3 or 4 were defined as unhealthy. 

The current treatment for disease was measured by participants’ responses to whether they were currently undergoing treatment for each of the diseases listed under Health Statistics on the Number of Elderly Medical Care Beneficiaries in Japan, with responses recorded ‘Yes’ or ‘No’.

##### Community Factors

We assumed that community factors included the participants’ sense of community, which was measured using the Community Commitment Scale (CCS) [38]. This scale consists of 8 items, 4 in each of the subscales on socializing and belonging. This was measured on a 4-point Likert scale, as follows: 0 = not confident at all, 1 = slightly unconfident, 2 = slightly confident, and 3 = fully confident. The total score ranges from 0 to 24. A high score indicate a high degree of community commitment. This scale has a Cronbach’s alpha of 0.75 among local volunteers in Japan [39]. Similar to HL, CCS scores were recorded for the overall CCS and the subscales on socializing and belonging.

### 2.4. Statistical Analysis 

We conducted an analysis of COCS-N score distribution by dividing it into three tertiles, designating the lower group as Q1, the middle group as Q2, and the higher group as Q3. Descriptive statistics were applied to the demographic data. To compare the three groups, we employed an analysis of covariance (ANCOVA). Multiple comparisons between these groups were carried out using Bonferroni’s test, which accommodates nonparametric tests. An ANCOVA was utilized to assess whether individual and community factors exhibited significant associations with CHWs’ abilities. In this study, an ANCOVA was used in the analysis because individual and community factors, which were considered to be factors that increase COCS-N among the three groups, were also scored higher by age and years of residence. Therefore, we decided to conduct the analysis using ANCOVA, which allowed us to adjust for these effects. Statistical significance was determined at a *p*-value of less than 0.05. We utilized IBM SPSS software version 24.0 (IBM Corp., Armonk, NY, USA) for data analysis.

### 2.5. Ethical Considerations

Participants were informed, both in writing and verbally, of the purpose and methods of the study, and that there would be no repercussions if they withdrew from or refused to participate in the study. They were informed that participation was voluntary and that completing and returning the questionnaire indicated their consent to participate in the study. This study was approved by the Ethical Review Board of Soka University (No. 2020-019).

## 3. Results

### 3.1. Demographic Characteristics

Table 1 shows the demographic characteristics of participants by groups. The mean overall age was 67.0 years (SD = 9.0 years). The prevalence of females was 88.8%. The most common family structure among all participants was living with a spouse, which accounted for 37.9% (Table 1).

### 3.2. Dependent Variable

Table 2 shows the COCS-N scores. The distribution of scores for all participants was checked, and Q1 and Q2 were divided by 14/15 points, while Q2 and Q3 were divided by 18/19 points. Of the 3120 participants, 1096 (35.1%) were in the Q1 group, 1035 (33.2%) were in the Q2 group and 989 (31.7%) were in the Q3 group. The mean COCS-N scores differed by five points between the Q1, Q2 and Q3 groups. The mean ± SD (range) of COCS-N scores was 16.2 ± 4.4 (0–24) for all participants, 11.5 ± 2.5 (0–14) for the Q1 group, 16.4 ± 1.1 (15–18) for the Q2 group, and 21.2 ± 1.7 (19–24) for the Q3 group (Table 2).

### 3.3. Demographic Characteristics by Group

Table 3 shows the demographic characteristics of participants by groups. The mean age was the Q1 group was 65.1 years (SD = 9.2 years), that of the Q2 group was 66.8 years (SD = 9.0 years), and the Q3 group’s was 69.2 years (SD = 8.4 years). The prevalence of females was the same for both Q1 and Q2 groups (89.8%); for Q3 group, it was 87.8%. The only significant differences were found for age (*p* < 0.001) and living arrangements (*p* = 0.004). There were no significant differences in sex (*p* = 0.376), education (*p* = 0.962), and years of residence (*p* = 0.439) (Table 3).

### 3.4. Independent Variables

Table 4 lists the related factors in the analysis of covariance (independent variables) of the COCS-N scores. As for individual factors, years of CHWs and HL were higher in the higher COCS-N score groups, with means ± SD of years of CHWs: 6.0 ± 6.0, 7.8 ± 7.0, and 8.2 ± 7.7, of All HLS-EU-Q47: 27.7 ± 6.6, 30.4 ± 6.9, 33.8 ± 7.8. Similarly, scores on the HLS-EU-Q47 HP, HLS-EU-Q47 DP, and HLS-EU-Q47 HC subscales of the HLS-EU-Q47 tended to be higher in the group with higher COCS-N scores. Individual factors related to significant differences in the COCS-N scores were years of community health work (F = 117.1, *p* < 0.001), and All HLS-EU-Q47 scores (F = 189.1, *p* < 0.001) and HLS-EU-Q47 HP (F = 151.1, *p* < 0.001), HLS-EU-Q47 DP (F = 95.1, *p* < 0.001), and HLS-EU-Q47 HC (F = 110.8, *p* < 0.001). As for community factors, All CCS scores were also higher in the higher COCS-N score groups, with means of 14.8 ± 3.7, 16.5 ± 3.5, and 18.2 ± 3.6. In addition, scores on the CCS subscale of CCS socializing and CCS belonging were similarly higher in the group with higher COCS-N scores. Community factors related to significant differences in COCS-N scores were overall CCS scores (F = 168.0, *p* < 0.001) and CCS socializing (F = 143.0, *p* < 0.001) and CCS belonging (F = 102.4, *p* < 0.001) (Table 4). 

## 4. Discussion

Our findings underscore a compelling positive correlation between COCS-N scores and specific determinants. Notably, “years of experience as a CHW” and “health literacy” in the individual domain, along with the “Sense of community” in the communal context, were firmly established as being significantly associated with CHWs’ competency. The clinical and policy implications of this study are related to strategies aimed at improving the competencies of CHWs, who act as a bridge between community residents and healthcare professionals, by examining relevant factors to ensure effective and equitable delivery of health care to the local population. The mean age of the study participants was 67.0 years (SD = 9.0 years) for all participants, 65.1 years (SD = 9.2 years) for the Q1 group, 66.8 years (SD = 9.0 years) for Q2 group, and 69.2 years (SD = 8.34 years) for Q3 group, while the prevalence of females was 88.8% in both the high and low COCS-N score groups. This value shows a trend identical to the characteristics of CHWs in Japan [40]. Therefore, it was inferred that the participants in this study were representative of CHWs in Japan.

The individual factor that was found to be significantly associated with COCS-N scores was years of experience as CHWs and Health literacy. For years of experience as CHWs, we found a relationship between the number of years of experience as CHWs and the COCS-N scores—the more the years of experience, the higher the COCS-N scores were. The results suggest that increasing the competencies shown in the COCS-N consisting of “Sharing the pleasure of living a healthy life” and “Creating healthy resources” will improve retention rates. For the next individual factor that was relevant, health literacy, the health literacy scores for the Q3 group with the highest COCS-N scores were as follows: All HLS-EU-Q47 33.8 (SD = 7.8), HLS-EU-Q47 HP 33.1 (8.6), HLS-EU-Q47 DP 37.3 (8.4), and HLS-EU-Q47 HC 31.3 (8.8). The health literacy scores for the Q1 group with the lowest COCS-N scores were as follows: All HLS-EU-Q47 27.7 (6.6), HLS-EU-Q47 HP 25.6 (7.5), HLS-EU-Q47 DP 31.9 (7.7), and HLS-EU-Q47 HC 25.9 (7.4). It is reported that the scores of the general Japanese population are lower than that of Europeans [37]. The average health literacy scores of the European population were as follows: All HLS-EU-Q47 33.8 (8.0), HLS-EU-Q47 HP 32.5 (9.1), HLS-EU-Q47 DP 34.2 (8.8), and HLS-EU-Q47 HC 34.7 (8.3). In comparison, the scores of the general Japanese population were as follows: All HLS-EU-Q47 25.3 (8.2), HLS-EU-Q47 HP 25.5 (9.2), HLS-EU-Q47 DP 22.7 (9.2), and HLS-EU-Q47 HC 25.7 (8.6) [37]. Additionally, the scores for older urban residents in Japan were as follows: All HLS-EU-Q47 27.5 (8.8), HLS-EU-Q47 HP 25.6 (10.2), HLS-EU-Q47 DP 31.2 (9.8), and HLS-EU-Q47 HC 26.0 (9.7) [39]. The health literacy scores for CHWs of Q1 group, with the lowest COCS-N scores, were higher than those of the general Japanese public. Furthermore, not only were the health literacy scores for the CHWs of Q3 group with the highest COCS-N scores higher than those of the general Japanese public, they were also equal to or higher than those of Europeans. There are two possible reasons for this. One is an access to and comprehension of health information: CHWs with higher health literacy have a superior ability to access and comprehend health-related information. This positions them to stay informed about the latest medical knowledge and health-related developments, which they can then effectively communicate to others, thereby enhancing their role as CHWs. The other is effective health communication skills: CHWs often need to communicate health information to community members. CHWs with high health literacy tend to possess superior communication skills, allowing them to convey health-related information effectively and in a comprehensible manner. This fosters a more impactful delivery of health promotion messages. The findings of this study underscore the importance of cultivating and harnessing health literacy to bolster CHWS efforts.

The community-related factor that was found to be significantly associated with COCS-N scores was a sense of community, which was measured by CCS scores. The results showed that the CHWs’ All CCS scores were 16.4 (SD = 3.9), with the Q3 group with the highest COCS-N scores 18.2 (SD = 3.6) and the Q1 group with the lowest COCS-N score, which was 14.8 (SD = 3.7). Previous studies found average CCS scores of 14.5 (SD = 4.1) for the general urban older population [41], and 13.0 (SD = 4.3) for mothers raising children [42]. Previous studies have reported that greater community attachment leads to more active participation in community activities [43], and higher trust in the community leads to stronger organizational capacity in the community [44]. Our findings support these studies, indicating that the greater the community commitment, the higher is the competence of CHWs. There are two possible reasons for this. One is cultural appropriateness: health-promotion workers with a high commitment to the community are more likely to adopt culturally appropriate approaches and exhibit cultural sensitivity. Cultural appropriateness is typically necessary to discuss when considering health care among immigrants and members of various ethnic minority groups [45,46]. However, because culture is a major determinant of lifestyle [47], the current study revealed that cultural appropriateness is necessary even in countries like Japan, where cultural divides appear to be small. This enables them to design and deliver health promotion programs that align with the unique culture and traditions of the community, making them more acceptable to community members. The other is the utilization of networks and resources: CHWs with a strong commitment to the community have easier access to community networks and resources and are more adept at building collaborative relationships. The role and function of CHWs includes social support and resource linkages [48], and in the process this will contribute to professional and community key-person-involvement, as well as community capacity building [49]. In the process of resource linking, CHWs also conduct community assessments, and the formation of such human–physical relationships is considered to enhance the sense of community [50]. This allows them to collaborate with other professionals and leaders to promote more comprehensive and effective health promotion initiatives. CHWs with a strong commitment to the community are expected to achieve more effective and sustainable outcomes in the design, implementation, and evaluation of health programs. Such commitment is an essential quality in CHWs’ efforts and is likely to have a significant impact on the health and well-being of the community.

### Limitations

This study has several limitations. First, the cross-sectional design meant that it was not possible to determine the causal relationships between COCS-N scores and individual- and community-related factors. Therefore, longitudinal and interventional studies are necessary to enable us to observe how CHWs’ activities and self-learning affect their competencies, and to examine how these change over time. Second, the participation rate of the study participants was low. Because the study was conducted during the COVID-19 epidemic, some Japanese CHWs were inactive in some municipalities. 

## 5. Conclusions

Our findings underscore a compelling positive correlation between COCS-N scores and specific determinants. Notably, “years of experience as a CHW” and “health literacy” in the individual domain, along with the “Sense of community” in the communal context, were firmly established as being significantly associated with CHWs’ competency. Consequently, the fortification of these individual- and community-related facets emerges as an efficacious strategy to enhance CHWs’ proficiency in NCD prevention, thereby further advancing public health.

## Figures and Tables

**Table 1 healthcare-12-00297-t001:** Demographic characteristics of participants.

	All (n = 3120)	
	n (%) or Mean (SD)	
Age	67.0	(9.0)
Sex		
Female	2777	(88.8)
Missing	11	(0.4)
Living arrangements		
Living alone	361	(11.6)
Living with spouse	1185	(37.9)
Living with children	184	(5.9)
Living with spouse and children	621	(20.0)
Living with spouse, children, and grandchild	185	(2.3)
Other	583	(22.2)
Missing	1	(0.0)
Education		
Primary and secondary schools	180	(5.8)
High schools	1505	(48.2)
Junior college or vocational schools	984	(31.5)
Universities	403	(12.9)
Graduate Schools	13	(0.4)
Other	10	(0.3)
Missing	25	(0.8)
Area of Residence		
Chubu	802	(25.7)
Kinki	685	(22.0)
Kanto	484	(15.5)
Tohoku	464	(14.9)
Kyusyu/Okinawa	396	(12.7)
Chugoku/Shikoku	240	(7.7)
Hokkaido	38	(1.2)
Missing	11	(0.4)
Years of residence	35.7	(16.1)

Those undergoing medical treatment were asked for multiple responses. SD: standard deviation.

**Table 2 healthcare-12-00297-t002:** Univariate analysis of COCS-N scores.

COCS-N Total	All (n = 3120)	Q1 (n = 1096)	Q2 (n = 1035)	Q3 (n = 989)
Mean (SD)	16.2 (4.4)	11.5 (2.5)	16.4 (1.1)	21.2 (1.7)
Median	16.0	12.0	16.0	21.0
Mode	16	14	18	24
Range	0–24	0–14	15–18	19–24
Skewness	−0.205	2.314	−1.249	0.281
Kurtosis	−0.268	−1.400	0.103	−0.268
**Perceptual Competence**	**All** **(n = 3120)**	**Q1** **(n = 1096)**	**Q2** **(n = 1035)**	**Q3** **(n = 989)**
Mean (SD)	9.13	6.97	9.25	11.39
Median	2.3	1.7	1.3	0.9
Mode	12	8	8	12
Range	0–12	0–12	6–12	7–12
Skewness	−0.448	−0.464	0.452	−1.566
Kurtosis	−0.297	0.980	−0.543	2.211
**Hehavioral Competence**	**All** **(n = 3120)**	**Q1** **(n = 1096)**	**Q2** **(n = 1035)**	**Q3** **(n = 989)**
Mean (SD)	7.05 (2.6)	4.49 (1.7)	7.18 (1.2)	9.75 (1.5)
Median	5.0	7.0	10.0	7.0
Mode	8	5	8	10
Range	0–12	0–8	3–11	7–12
Skewness	−0.574	−0.316	−0.014	−0.152
Kurtosis	0.348	0.308	−0.990	−0.275

SD: standard deviation, COCS-N Total Scores Q1: a group with COCS-N score range of 0 to 14; Q2: 15 to 18; Q3: 19 to 24, perceptual competence scores Q1: a group with COCS-N score range of 0 to 12; Q2: 6 to 12; Q3: 7 to 12. Behavioral competence scores Q1: a group with COCS-N score range of 0 to 8; Q2: 3 to 11; Q3: 7 to 12.

**Table 3 healthcare-12-00297-t003:** Demographic characteristics of participants by group.

	Q1 (n = 1096)		Q2 (n = 1035)		Q3 (n = 989)		F-Value (*p*-Value) or χ^2^-Test
	n (%) or Mean (SD)		n (%) or Mean (SD)		n (%) or Mean (SD)		
Age	65.1	(9.2)	66.8	(9.0)	69.2	(8.4)	54.0 (*p* < 0.001)
Sex							
Female	980	(89.4)	929	(89.8)	868	(87.8)	0.98 (*p* = 0.376)
Male	6	(0.5)	3	(0.3)	2	(0.2)	
Living arrangements							
Living alone	102	(9.3)	136	(13.1)	123	(12.4)	*p* = 0.004
Living with spouse	393	(35.9)	386	(37.3)	406	(41.1)	
Living with children	59	(5.4)	63	(6.0)	62	(6.3)	
Living with spouse and children	248	(22.6)	202	(19.5)	171	(17.3)	
Living with spouse, children, and grandchild	72	(6.6)	50	(4.8)	63	(6.4)	
Other	221	(20.2)	198	(19.1)	164	(16.6)	
Missing	1	(0.1)	0	(0.0)	0	(0.0)	
Education							
Primary and secondary schools	66	(6.0)	53	(5.1)	61	(6.2)	*p* = 0.962
High schools	521	(47.5)	506	(48.9)	478	(48.3)	
Junior college or vocational schools	344	(31.3)	332	(32.1)	308	(31.1)	
Universities	149	(13.6)	124	(12.0)	130	(13.1)	
Graduate Schools	4	(0.4)	5	(0.5)	4	(0.4)	
Other	4	(0.4)	4	(0.4)	2	(0.2)	
Missing	8	(0.7)	11	(1.1)	6	(0.6)	
Years of residence	34.5	(16.3)	35.6	(16.1)	37.1	(16.0)	0.82 (*p* = 0.439)

Those undergoing medical treatment were asked for multiple responses. SD: standard deviation, Q1: a group with COCS-N score range of 0 to 14, Q2: a group with COCS-N score range of 15 to 18, Q3: a group with COCS-N score range of 19 to 24.

**Table 4 healthcare-12-00297-t004:** Results of tests for differences between groups of factors related to COCS-N scores.

						n = 3120
	Q1		Q2		Q3	F-Value (*p*-Value)
	1096 (35.1)		1035 (33.2)		989 (31.7)	
Individual factors
Years of community health work	6.0 ± 6.0		7.8 ± 7.0		8.2 ± 7.7	117.1 (*p* < 0.001)
	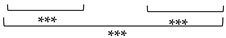	
All HLS-EU-Q47 *	27.7 ± 6.6		30.4 ± 6.9		33.8 ± 7.8	189.1 (*p* < 0.001)
	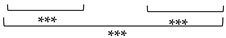	
HLS-EU-Q47 HP *	25.6 ± 7.5		29.2 ± 7.7		33.1 ± 8.6	151.5 (*p* < 0.001)
	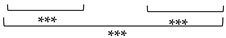	
HLS-EU-Q47 DP *	31.9 ± 7.7		34.1 ± 7.9		37.3 ± 8.4	95.1 (*p* < 0.001)
	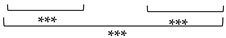	
HLS-EU-Q47 HC *	25.9 ± 7.4		28.2 ± 7.6		31.3 ± 8.8	110.8 (*p* < 0.001)
	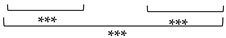	
Community factors
All CCS *	14.8 ± 3.7		16.5 ± 3.5		18.2 ± 3.6	168.0 (*p* < 0.001)
	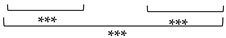	
CCS Socializing *	7.4 ± 2.1		8.3 ± 2.0		9.3 ± 2.1	143.0 (*p* < 0.001)
	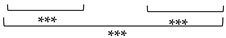	
CCS Belonging *	7.3 ± 2.2		8.2 ± 2.0		8.1 ± 2.3	102.4 (*p* < 0.001)
	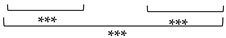	

***: *p* < 0.001, Years adjusted. Years of community health work (years). *: All HLS-EU-Q47 (0.0–50.0). HLS-EU-Q47 HP (0.0–50.0). HLS-EU-Q47 DP (0.0–50.0). HLS-EU-Q47 HC (0.0–50.0). All CCS (8.0–32.0). CCS Socializing (4.0–16.0). CCS Belonging (4.0–16.0).

## Data Availability

The data that support the findings of this study are available from Yokohama City University, but restrictions apply to the availability of these data under the Japan Personal Information Protection Law, since they were used under license for the current study, and so are not publicly available. Data are, however, available from the first/corresponding authors upon reasonable request and with the permission of Yokohama City University.

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
