# Peer review of "Empowering Community Health Workers in Japan: Determinants of Non-Communicable Disease Prevention Competency"

_healthcare, 2024, doi:10.3390/healthcare12030297_

Round 1

Reviewer 1 Report

Comments and Suggestions for Authors

Thank you for the opportunity to review this paper.

The paper aims at assessing the aptitude of Community Health Workers (CHWs) in Non-communicable Diseases (NCDs) prevention by introducing the Community Health Workers Perceptual and Behavioral Competency 20 Scale for preventing non-communicable diseases (COCS-N). The ultimate goal is to empower CHWs with competencies helpful for their effective performance of their roles in the communities they serve.

The introduction is written well and gives a strong basis for the importance of CHWs in health care provision within their communities through various strategies. However, it is not clear:

(i)               What ‘empowerment’ means in general or how it is operationalized in this study.

(ii)              What ‘competency’ means in general or how it is operationalized in this study.

The two terms are used but not defined.

This was a cross-sectional design as a secondary analysis using data from self-administered questionnaire survey from 6,480 CHWs residing in municipalities across all Japan. A total of 3,651 individuals were included in this study (response rate: 56.3%). A further 531 individuals were excluded for not participating in the health promotion activities for a sufficient duration or did not respond to all the items in the competency scale or failed to answer at least two questions on the European Health Literacy Scale. his left 3,120 participants for inclusion in the analysis, a valid response rate of 48.1%. According to the authors the low response rate was particularly so because the data was collected during the COVID-19 pandemic.

Appropriate statistical analysis techniques were employed in this study. Besides descriptive analysis, ANCOVA was run for comparing differences in group means, and  Bonferroni's test, which accommodates nonparametric tests was used for multiple comparisons between the groups.

Results are well presented.

However, I would have suggested the information in Table 2 comes first (as Table 1) and vice versa because often we deal with descriptive information before moving on.

In the demographic results, (i) why is location/residence not included yet the study encompasses rural as well as urban areas?  (ii) Employer status – Private, NGO, Government is not stated yet this determines the remuneration level as well as resources available.

The rest of the paper is very systematic in the flow of information making it possible for the reader to understand the topic. The discussion and conclusions are supported with the findings of the study.

The study provides useful findings for practical applications not just in Japan but the competency framework as well as the health literacy scale could easily be adopted in other countries and tailored to contextual needs.

Reviewer 2 Report

Comments and Suggestions for Authors

1.          The method description in the abstract could be more concise, providing only essential details. However, the results section requires more elaboration to provide a comprehensive overview.

2.          The characteristics of Community Health Workers (CHWs) vary across countries. It is suggested that the authors provide more context on the significance of CHWs in Japan and their role in addressing NCDs.

3.          COCS-N is divided into three groups. Please provide the score ranges for each of the three groups.

4.          The statistical analysis employed ANCOVA, and the rationale behind this was not explained in the statistical analysis section. It is observed from Table 3 that the adjustment was made for Years & Years of CHWs. It is recommended to provide additional clarification on this adjustment in the statistical analysis section.

5.          The demographic characteristics in Table 2 should be analyzed using appropriate statistical tests to examine the differences among the three COCS-N groups.

6.          The COCS-N includes perceptual and behavioral subscales, yet the statistical results in this manuscript only present the total score. It is recommended that the authors incorporate an analysis of the subscales to provide a more detailed understanding of the factors involved and to support the points made in the discussion.

7.          The authors suggested two potential reasons for the positive correlation between a greater of community commitment and higher competences among community health workers. However, there was no citation of any literature to support these points. It is recommended to include relevant references to enhance the evidential support for the argument.

Reviewer 3 Report

Comments and Suggestions for Authors

Dear authors, I want to express my gratitude for giving me the chance to review your work. I believe that your writing has a lot of potential, and I would like to offer some constructive feedback to help you improve it even further.

The article discusses an important topic in the field of Nursing, especially in Japan. This paper focus the empowerment of Community Health Workers, regarding the Non Communicable Disease Prevention competency.

Based on my assessment, I have identified certain areas where the paper could be improved. However, after implementing my suggestions, I am confident that the article is ready for publication.

The title of the article does not answer the objective of the study.

Abstract

The abstract provides a clear understanding of the study's aim and the questionnaires used, but it lacks information about the period during which the study was conducted. This information is crucial for contextualizing the study's relevance and scope. Additionally, the abstract contains numerous abbreviations that could potentially complicate the reader's comprehension. Reducing the number of abbreviations can improve clarity.

Furthermore, the abstract could benefit from emphasizing the practical applications of the research findings, particularly in clinical practice and the development of health policies. By highlighting the real-world implications of the study, readers can better understand its significance and potential impact. Additionally, the abstract could be more concise, avoiding an excessive amount of detailed information that may overwhelm the reader and obscure the main message.

 Keywords

The keywords selected are related to the study but don't correspond to indexed / Mesh terms.

Introduction/literature review

The title of the article does not answer the objective of the study. The concept of empowering is not explored in the introduction.

Line 65-67 which authors or studies support these conclusions and state that determining appropriate compensation for their efforts is a significant challenge. Supporting this statement could help reinforce the relevance of this study.

The objective and contributions are clear (line 122-125).

The literature review is included in the introduction, it could be a different chapter.

Although the definition of central concepts is clear.

Methods

The chapter of methodology is well structured and the sub-chapters are clear.

- The scales are well-grounded and appropriate to the context.

- How was the anonymity and confidentiality of the data guaranteed?

Results

The results are well structured. Although the text sometimes is confusing, the tables help to clear their presentation.

Discussion and conclusion

They are well articulated and have adequate sub-chapters, which help to maintain the thread while reading the article. The ideas have been expressed in a very articulate and clear manner. But some aspects need to be reviewed:

- Line 285, you refer twice the mean age for Q2. Is it one of them about Q3?

- In line 287, you mentioned that the results are consistent with the characteristics of CHW in Japan. However, these characteristics haven't been clearly described in the introduction. Therefore, it would be beneficial to provide a more detailed explanation of these characteristics to make the discussion more objective.

- Line 296: It is not a comma after literacy, but a period;

The acknowledgment of the study's limitations is relevant and is described clearly and transparently.

The recommendation to develop additional studies on the subject is very important, encouraging other researchers to continue studying the subject.

References

The cited references are recent, the majority of them, from the last 5 years, although some are from the last 10 years. The references previous to this time-space are from the framework and from the scales used in this study, giving substance.

I would like to congratulate the authors on this important paper.

Round 2

Reviewer 1 Report

Comments and Suggestions for Authors

I want to appreciate your detailed explanation on the revisions made on your manuscript.

I have read the revised version of the paper alongside the letter from the authors. All my prior concerns/comments have been adequately addressed. The paper has good and helpful information relevant for policy implications in CHWs programs (trainings), and interventions (implementation) that they participate in. 

Reviewer 2 Report

Comments and Suggestions for Authors

The manuscript has been appropriately revised, and there are no further comments.

Congratulations to the authors on the acceptance of their publication.

Reviewer 3 Report

Comments and Suggestions for Authors

Dear authors, I want to take a moment to express my gratitude for your openness to feedback and willingness to make necessary changes to the article. It's clear that you have put a great deal of time and effort into creating a valuable piece of work. After carefully reviewing the revised article, I feel confident in saying that it is now ready for publication. Congratulations on this accomplishment.